Customizable pattern synthesis: a deep generative approach for lantern designs

Yan Mengran 1 2 17556985866@163.com
Tang Chun 1
Yan Jida 3
Surip Siti Suhaily 2 15551583666@163.com
1 Fine Arts Department, Bozhou University , Bozhou City, Anhui Province , China
2 Product Design Department, School of The Arts, Universiti Sains Malaysia , Penang , Malaysia
3 International Education Institute, Quanzhou University of Information Engineering , Quanzhou City, Fujian Province , China
Do Trang
Electronic publication date: 2025 Mar 7
Publication date: 2025
Volume: 11
Electronic Location ID: e2732
Received 2024 Dec 11; Accepted 2025 Feb 5
Copyright: © 2025 Yan et al.
Copyright year: 2025
Copyright holder: Yan et al.
License: This is an open access article distributed under the terms of the Creative Commons Attribution License, which permits unrestricted use, distribution, reproduction and adaptation in any medium and for any purpose provided that it is properly attributed. For attribution, the original author(s), title, publication source (PeerJ Computer Science) and either DOI or URL of the article must be cited.
License URL: https://creativecommons.org/licenses/by/4.0/

Keywords: Lantern patterns, Deep learning, Generative model, Pattern synthesis

Funding: The authors received no funding for this work.

==============================
Pattern design is essential in various domains, especially in traditional lantern production, where patterns convey cultural history and artistic values. Our research presents an innovative generative model that produces customizable lantern patterns, integrating classical aesthetics with modern design features via a generative adversarial network (GAN)-based framework. The model was trained on an extensive dataset of over 17,000 pattern images over ten various categories. Experimental assessment demonstrates the model’s remarkable proficiency, achieving an Inception Score of 5.259, much surpassing the performance of other GAN-based approaches. This exceptional result demonstrates the effective integration of traditional pattern elements with AI-driven design processes. The model offers enhanced design flexibility via noise vector hybridization and post-processing techniques, allowing for accurate control over pattern production while preserving cultural authenticity. These capabilities make our model a valuable tool for modernizing lantern pattern design while maintaining classic artistic elements.

Introduction

Essence of pattern design

The roles of pattern design are significant across various fields, especially in architecture, fashion, decoration, and children’s toy production. Different creative fields use patterns to make their work more meaningful and beautiful. When artists, designers, or craftspeople add patterns, they can express their culture and creativity while also making their work look more polished and professional. Think of how a quilt pattern both looks attractive and tells a story about its maker’s traditions (Alencar, Cowan & Lucena, 1996). In architecture, pattern design provides a framework for creating functional and aesthetically pleasing structures. Architectural patterns help architects balance different needs, enabling them to make thoughtful design choices that consider usability and sustainability (Harrison & Avgeriou, 2007). Islamic architecture often incorporates geometric patterns into its designs. These patterns showcase how mathematical principles can create beautiful, culturally significant art that stands the test of time (Sobh & Samy, 2018). Selecting suitable pattern designs not only elevates the visual appeal of structures but also ensures they effectively meet the practical requirements of their surroundings (Eden & Kazman, 2003). In the fashion industry, pattern design is essential for creating cohesive and visually appealing collections (Liu et al., 2019). Patterns in fashion can range from prints and textures to structural elements of garments. They serve as a means of expression and identity, reflecting cultural influences and trends (Yang, 2017). The use of traditional decorative symbols illustrates how patterns can convey deep cultural meanings and societal values, influencing consumer behavior and preferences (Appiah-Kubi et al., 2023). This intersection of design and cultural significance underscores the importance of patterns in fashion as a medium for storytelling and identity formation (McLean, 2008). Decorative patterns, whether in interior or product design, create harmonious and inviting spaces. These patterns transform environments, making items both functional and visually appealing (Ochoa et al., 2002). Additionally, using patterns in decoration influences people’s emotional responses to a space. Moreover, pattern design plays a crucial role in creating children’s toys by addressing safety concerns (Ismail et al., 2019), educational value (Abdi & Cavus, 2019; Saikia, Bhattacharyya & Baruah, 2023), gender considerations (Cherney, 2018), and sustainability (Feng, 2024). By integrating these elements into the design process, toy manufacturers can produce products that positively contribute to children’s growth and development (Feng, 2024). The ongoing evolution of pattern design in this field reflects a commitment to improving the quality of children’s play experiences.

Pattern design of lanterns

First created in the Eastern Han Dynasty as light sources, these lanterns represent joy, positive fortune, and social status in Chinese culture. However, advancements in lighting technology and a wider range of decorative choices have reduced their use (Ming, Mohd Nasseri & Husain, 2023). Today, lantern designs depend on various factors, such as cultural meaning, art styles, and materials. Understanding these factors helps create lanterns that are both functional while still reflecting the combination of traditional and modern art styles. Chinese lantern designs are strongly influenced by a rich cultural heritage rooted in traditional patterns. These patterns have been developed and accumulated over thousands of years and now play a central role in lantern design. Zhao & Sahari (2024) points out that these traditional patterns are living heritage that attracts people and motivates young adults to learn more about history. Similarly, Yu & Ghazali (2023) emphasizes the value of artistic symbols in design, as they can tell rich stories and express cultural identities, thereby enriching the uniqueness of products. The studies highlight the need to balance preserving cultural heritage and promoting innovative design. This balance provides valuable insights for global cultural exchange and the creative industries’ growth (Yu & Ghazali, 2023; Zhao & Sahari, 2024). Ming, Mohd Nasseri & Husain (2023) examined the declining popularity of traditional Chinese lanterns among young Malaysian Chinese, explaining their reduced interest and suggesting novel approaches to developing new cultural products.

Generative models in pattern design

Generative models have recently been applied to pattern design, and this trend is expected to grow in popularity as they produce detailed and innovative designs based on learned patterns. These models can be developed using various algorithms, from genetic programming (Zhu & Gao, 2022) to deep learning (Goodfellow et al., 2014; Kingma & Welling, 2013). As a result, the design process is significantly shortened with the facilitation of these approaches. Zhu & Gao (2022) emphasized the importance of incorporating traditional cultural elements into modern packaging to enhance both visual appeal and cultural relevance. They introduced a method that employs genetic algorithms (GAs) to create varied pattern designs by mimicking natural selection and evolutionary processes. This technique allows designers to efficiently explore numerous design options, resulting in innovative packaging that resonates with cultural themes. Their findings indicate that GAs can successfully balance aesthetic goals with functional limitations, providing a valuable approach for designers seeking to blend traditional motifs with modern packaging concepts (Zhu & Gao, 2022). Singh, Kumar & Sharma (2013) developed a generative model for parametric pattern design. This model identifies and creates decorative patterns that focus on aesthetics, enhancing a product’s visual and emotional appeal by analyzing shapes and spatial interactions among motifs. It allows designers to explore variations in design patterns and produce intricate, culturally rich designs. Combining traditional artistic elements with computational tools offers strong support for designers (Singh, Kumar & Sharma, 2013). Moreover, our methods could be applied to generate novel designs for diverse cultural objects beyond lanterns. The model’s capabilities suggest it could assist in conceptualizing new products, architectural renderings, or artistic creations across design industries, such as furniture, apparel, or consumer electronics.

Motivations

Recently, generative models developed with deep learning have revolutionized decorative pattern design. Various architectures, such as generative adversarial network (GAN) (Goodfellow et al., 2014) and variational autoencoder (VAE) (Kingma & Welling, 2013), have been widely used to generate high-quality designs. In textile and architectural design, these models can automate complex pattern generation while maintaining aesthetic appeal (Ahteck et al., 2024). They allow users to blend traditional motifs with modern elements, producing patterns that appeal to contemporary style while retaining cultural significance (Abdel Halim, Ibrahim & Abdel Tawab, 2024). The application of artificial intelligence (AI)-driven generative models in product packaging further demonstrates their utility as companies integrate cultural themes into pattern designs to engage consumers more effectively (Wang, 2024). However, these existing models face challenges related to data quality and model interpretability, as high-quality, diverse datasets are essential for generating coherent and meaningful patterns (Regenwetter, Nobari & Ahmed, 2022).

The proposed generative model has broader applicability beyond the specific use case of lantern design. Its ability to generate diverse, high-quality data, including synthetic images (Hussain et al., 2020), indicates its potential value across various design industries. Furthermore, the model could be used to assist product designers in conceptualizing new furniture, apparel, or consumer electronics by generating novel design ideas. Similarly, the model’s capabilities could benefit architectural design by enabling the rapid creation of realistic 3D building renderings. The model is expected to produce synthetic images that closely resemble real-world visual data, making it well-suited for such design applications. Therefore, the development of image generative models can provide a versatile tool that could enhance creative processes across many design-focused industries.

Dataset for modeling

To train our generative model, we used a dataset collected by Yar et al. (2023). The dataset contains 17,433 samples, each belonging to one of ten categories: Abstract, Cartoons, Cheetah, Ethnic, Geometric, House, Leaves, Paisley, Seamless, and Tech & Food patterns. Among these categories, House and Abstract patterns account for a relatively small portion of the dataset, with 272 and 283 samples, respectively. The Seamless pattern, on the other hand, has 5,711 samples, making up 32.7% of the total. For the other patterns, the sample counts range from about 600 to 3,000. Before training, we resized all images to a uniform size of 64×65×3. The dataset was then randomly split into a training set and a test set with a stratified ratio of 9:1. The training set contains 15,690 samples, while the test set has 1,743 samples. Table 1 summarizes the numbers of samples in each category of the used dataset.

Table 1 Numbers of samples in each category of the used dataset.

No.	Category pattern	Number of samples	
1	Abstract	283	
2	Cartoons	1,470	
3	Cheetah	598	
4	Ethnic	2,512	
5	Geometric	908	
6	House	272	
7	Leaves	3,267	
8	Paisley	952	
9	Seamless	5,711	
10	Tech & food	1,460	
Total	17,433	

Generative model

Model architecture

Figure 1 shows the architecture of our generative model with two networks that are adversarially trained. This architecture is based on the generative adversarial network (GAN) design (Goodfellow et al., 2014), with modified layers specifically tailored to address this issue. Our model has two separate networks: an Encoder and a Decoder. The Encoder is designed with a Linear layer and three 2-Dimensional Convolutional (Conv2D) layers. First, a noise vector of size 1 × 100 is randomly sampled from a normal distribution N(μ=0,σ=1). The linear layer expands the noise vector’s size to 1 × 8,192, which is then reshaped to create a matrix of size 4×4×512. These three Conv2D layers reduce the output channels from 512 to 3 and increase the matrix’s dimensions from 4×4 to 64×64, creating pattern images of size 64×64×3. The Decoder is designed with three Conv2D layers and a linear layer. Both generated and real images of size 64×64×3 are inputs to the Decoder. The three Conv2D layers increase the image depth from 3 to 512 before reducing their dimensions to 8×8. This matrix of size 8×8×512 is then flattened and processed by the Linear layer to predict whether the input image is generated or real.

Figure 1 Model architecture used for pattern image generation.

The model has two networks: Encoder and Decoder. The Encoder learns to generate pattern images while the Decoder is trained to effectively distinguish between real pattern images and generated ones. The equilibrium point is reached once the Decoder is unable to distinguish between two classes of pattern images.

Generative adversarial network

GAN is one of the most effective designs for generative modeling that was first proposed by Goodfellow et al. (2014). A fundamental GAN-based generative model has two networks: the Generator (GEN) and the Discriminator (DEC) (Fig. 2) that involves in an adversarial competition. The Generator is trained to create synthetic data, while the Discriminator is trained to detect whether the data are synthetic or real. The Generator progressively enhances the quality of the synthetic data during training each time it receives feedback from the Discriminator. Training continues until an equilibrium is reached. At this theoretical equilibrium, the Generator is able to produce synthetic data that the Discriminator cannot reliably distinguish from real data.

Figure 2 Fundamental design of of generative adversarial network.

In the context of mathematical expressions, the Discriminator learns to approximate the distribution preal(x) of the real samples x, while the Generator learns to produce synthetic samples from noise vectors z. These noise vectors are randomly sampled from a prior distribution psynthetic(z). The goal of the Encoder’s training is to bring the real sample distribution x∼preal(x) closer to the synthetic sample distribution GEN(z)∼psynthetic(z). The objective of the Discriminator’s training is to enhance its capacity to distinguish real samples from synthetic ones. The loss function for the Discriminator is given by:

(1) maxDEC⁡V(DEC)=Ex∼preal(x)[logDEC(x)]+Ez∼psynthetic(z)[log⁡(1−DEC(GEN(z)))],

where Ex∼preal(x)[logDEC(x)] and Ez∼psynthetic(z)[log⁡(1−DEC(GEN(z)))] are expectations for detecting real and synthetic samples, respectively. The Generator aims to improve the quality of synthetic samples to challenge the Discriminator with the loss function is given by:

(2) minGEN⁡V(GEN)=Ez∼psynthetic(z)[log⁡(1−DEC(GEN(z)))].

By combining the loss functions of both networks above, the loss function of GAN is formulated as:

(3) minGEN⁡maxDEC⁡V(DEC,GEN)=Ex∼preal(x)[logDEC(x)]+Ez∼psynthetic(z)[log⁡(1−DEC(GEN(z)))].

The adversarial training of these two networks is terminated once the two distributions, preal(x) and psynthetic(z), reach a point of equivalence, allowing any noise vector z randomly sampled from psynthetic(z) to produce new samples GEN(z). However, in practice, training GAN-based models is highly challenging because the Discriminator tends to learn more quickly than the Generator. This situation can cause mode collapse, a condition where the Generator outputs only low-variation samples. To address this problem, we customized our model based on Wasserstein GAN (WGAN), a variant of GAN that uses Earth-Mover’s distance to quantify distribution differences for training. The original WGAN architecture was introduced by Arjovsky, Chintala & Bottou (2017). WGAN approaches like ours provide more stable measures of distribution differences compared to traditional GANs (Wang & Nguyen, 2025a). By introducing a critic network with gradient clipping, WGAN approaches reduce mode collapse and training instability, allowing for more consistent learning of complex distributions. Unlike previous approaches, WGANs offer a mathematically robust framework that improves generative modeling across domains like image generation and style transfer.

Model training

We trained our model for 400 epochs with a learning rate of 1.5×10−3, utilizing the Adam optimizer (Kingma & Ba, 2014) for efficient gradient-based optimization. The optimizer’s default parameters, β1=0.9 and β2=0.999, were used to balance the convergence rate and stability during the training process. To prevent potential overfitting, early stopping was applied by monitoring changes in the training loss of the Encoder and Decoder. The training process was terminated if no improvement was observed in either network for 20 consecutive epochs.

Evaluation metric

The Inception Score (IS) is a widely recognized metric used to evaluate the performance of GAN-based generative models, particularly regarding the quality and diversity of generated images. This metric, first introduced by Salimans et al. (2016), leverages a pre-trained Inception model to assess the generated images. The Inception model is a pre-trained deep convolutional neural network developed using the ImageNet dataset (Szegedy et al., 2015). The score is calculated based on the distribution of predicted class labels for the generated images, measuring both the clarity of the images and the diversity of the classes represented. It is mathematically defined as:

(4) IS=exp⁡(Ex∼psyntheticDKL(p(y|x)||p(y))),

where p(y|x) is the conditional probability of class labels given an image, p(y) is the marginal distribution of class labels across all generated images, and DKL(p(y|x)||p(y)) is the Kullback-Leibler Divergence between p(y|x) and p(y) (Wang & Nguyen, 2025b). Higher IS indicate that generative models can produce images of higher quality and diversity.

Computing resources

The training process was carried out using PyTorch 2.0 on an RTX 3080 GPU with 12 GB of memory. All computations were performed on a Windows 11 system featuring an AMD Ryzen 5 5800X 8-Core Processor (3.80 GHz) and 16 GB of RAM.

Results and discussion

Model evaluation

Besides our proposed methods, we also implemented two other GAN-based models for comparison: Wasserstein GAN (WGAN) (Arjovsky, Chintala & Bottou, 2017) and Least Squares GAN (LSGAN) (Mao et al., 2016). All the generative models were trained on the same dataset for 400 epochs. After completing the training, the models were used to generate synthetic images. By assessing the quality and diversity of the synthetic pattern images, the overall performance of the generative models was evaluated. Table 2 provides information on the Inception Score (IS) and Fréchet Inception Distance (FID) for the synthetic pattern images. The results indicate that the WGAN model achieved the lowest IS of 2.123, while the LSGAN model obtained an IS value of 3.298. Compared to these models, our proposed method demonstrated superior performance with an IS value of 5.259. This high IS value highlights that our model produces synthetic images with significantly higher quality and diversity than those generated by the other models. FID measures the distance between the real and synthetic image distributions, with lower values indicating better performance. The table shows that our proposed model has the lowest FID of 20.1, compared to 64.5 for WGAN, 45.2 for LSGAN, and 38.9 for WGAN-GP. This indicates that the synthetic images generated by our model are closer to the real image distribution, further demonstrating the superior performance of our proposed method. In summary, the comparative study indicates that our proposed generative model outperforms the other GAN-based models on both measures of image quality and diversity. Figure 3 visualizes a small set of 30 synthetic images produced by our model.

Table 2 Performance of our generative model compared to other generative models.

No.	Method	Inception score	Frechet inception distance	Training time (s)	
1	WGAN	2.123	64.5	815	
2	LSGAN	3.298	45.2	528	
3	WGAN-GP	3.572	38.9	965	
4	Ours	5.259	20.1	411	

Figure 3 The synthetic pattern images produced by the proposed generative model.

Method’s limitations

Our generative model as well as other GAN-based models face several challenges that may limit their effectiveness. A major issue is mode collapse, where the generator produces a limited range of outputs, failing to capture the full diversity of the training data (Wang et al., 2024). The adversarial dynamics between the Generator and Discriminator often lead to unstable training and poor convergence, especially vanishing gradients. Additionally, our model typically requires large datasets, which is particularly restrictive in fields with limited data availability, like medical applications. Furthermore, it struggles with output quality and interpretability, as synthetic images can contain artifacts or distortions and lack meaningful representation of the intended concepts.

Besides, our modeling process remains biases that could substantially impact the generative model’s performance, including severe category imbalances (with Seamless patterns comprising 32.7% and some categories having fewer than 300 samples) and potential cultural and sampling limitations. These biases may lead to skewed pattern generation, potentially reinforcing existing design stereotypes and reducing the model’s true creative potential across diverse cultural contexts. The methodological constraints, such as training on a relatively small dataset of 17,433 images and relying on a single computational setup, further compound these potential biases. Consequently, the findings might not comprehensively represent the full complexity of global pattern design traditions, suggesting a need for expanded data sources, multiple evaluation metrics, and cross-cultural validation.

Pattern customization

Our generative model can be combined with multiple approaches for customizing pattern generation specifically designed for lattice designs and decorative elements. The key customization technique utilizes noise vector hybridization, which combines randomly generated noise with extracted features from existing pattern images using linear transformation matrices. This method enables the preservation of desired pattern characteristics while generating novel variations. Users can also incorporate conditional parameter inputs, where specific design requirements can be encoded to influence the generation process. The model facilitates latent space interpolation between noise vectors, creating smooth transitions across different pattern variations. Post-generation refinements can be applied through standard image processing techniques to adjust colors, enforce symmetry rules, or create seamless pattern repetitions. These integrated customization methods give designers precise control over pattern attributes while ensuring output diversity and coherence. Figure 4 illustrates two customization methods with their major steps for lantern design.

Figure 4 Pattern customization for lantern design.

Pattern generation is driven by either (i) random noise in combination with concept image or (ii) random noise only.

Conclusion

In our study, we proposed a GAN-based generative model specifically designed to produce pattern images for lantern design. The results demonstrate that our model successfully generates high-quality images featuring a wide diversity of patterns, enabling a broader range of aesthetic and thematic customization options. Compared to other generative models, our GAN-based approach achieved the highest Inception Score, indicating a strong ability to produce images of high quality and variety. Generative models like ours play essential roles in creating visually appealing patterns that meet decorative criteria in terms of structural integrity and complexity. Applying artificial intelligence supports artists in their creative process, enabling them to design patterns that align with various cultural or modern aesthetics. However, our approach confronts several critical limitations that warrant careful consideration and future research. The model’s scalability is constrained by its reliance on large, balanced datasets, with current performance demonstrating significant variations across pattern categories. Generalizability remains a challenge, as the model struggles with mode collapse and producing diverse outputs, particularly for underrepresented pattern types.

Additional Information and Declarations

Competing Interests

The authors declare that they have no competing interests.

Author Contributions

Mengran Yan conceived and designed the experiments, performed the experiments, analyzed the data, performed the computation work, prepared figures and/or tables, authored or reviewed drafts of the article, and approved the final draft.

Chun Tang conceived and designed the experiments, performed the experiments, analyzed the data, authored or reviewed drafts of the article, and approved the final draft.

Jida Yan conceived and designed the experiments, performed the experiments, analyzed the data, authored or reviewed drafts of the article, and approved the final draft.

Siti Suhaily Surip conceived and designed the experiments, performed the experiments, analyzed the data, authored or reviewed drafts of the article, and approved the final draft.

Data Availability

The following information was supplied regarding data availability:

The code used in the experiments is available at Zenodo: Yan, M. (2025). Customizable pattern synthesis. Zenodo. https://doi.org/10.5281/zenodo.14767932.

The data is sourced from Yar et al. (2023) and is available at GitHub: https://github.com/Gnahy/TexGAN.

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
