# Peer review of "Customizable pattern synthesis: a deep generative approach for lantern designs"

_PeerJ Computer Science, doi:10.7717/peerj-cs.2732_

## Round 0.1 · original submission · Major Revisions

Please consider the comments from the reviewers and revise the manuscript accordingly. Also, consider incorporating more experiments, such as testing the method by Ishaan et al. as suggested by Reviewer 2; this would enhance the comprehensiveness of the study.

Reviewer 1 ·

Basic reporting

-The manuscript is well-structured but requires adjustment to enhance clarity. Some sentences in the introduction and results sections are overly complex and should be rewritten.
(a) Line 27–29 "Each field utilizes patterns to add value to items or objects, display creativity, cultural aspects, and rituals, and emphasize perspectives on artistic processes, thereby enhancing the overall quality of the output"
(b) Line 32–34 "Furthermore, the integration of geometric patterns, particularly in Islamic architecture, demonstrates how mathematical principles can produce timeless designs imbued with cultural significance"
(c) Line 63–65 "Both studies underscore the importance of preserving cultural heritage while promoting innovative design to provide insights for global cultural exchanges and the development of creative industries"

- The background section is informative but lacks detail. The explanation of Wasserstein GANs and their applications is insufficient. More information is needed to clarify how this study differs from previous work.
- Figure 4 is confusing, it needs better labels to show how the customization works.
- Figure 3 is unclear and needs to improve resolution for better evaluation.
- Table 2 doesn't provide enough details and should be expanded with more information.

Experimental design

- The paper does not adequately explain the unique contributions of its GAN model. The description of the model and training process lacks information like hyperparameters and training duration
- The evaluation only relies on the Inception Score and provides insufficient comparisons to other GANs. Including additional metrics and visual comparisons would improve the analysis.

Validity of the findings

- The study’s focus on lanterns limits its broader relevance. Additionally, it’s unclear how significant this work is compared to other GAN applications in design. The authors should explain how their methods could be used for other kinds of designs or cultural objects.
- Details about the dataset (origin, potential biases, etc) are insufficient. To make the results more reliable, the authors need to discuss any potential biases in their data and explain how these biases might affect their findings.
- The conclusions are reasonable but need to address the method's limitations. The authors need to add a discussion of the method’s limitations, such as scalability, and generalizability, and suggestions for future research.

Cite this review as

Reviewer 2 ·

Basic reporting

- The paper is written clearly and uses professional English throughout, ensuring it is easy to follow and unambiguous.
- While the study provides sufficient background and context, it overlooks some related works, which are highlighted below.
- The figures are of high quality, and both the data and code have been shared appropriately.

Experimental design

- The research aligns well with the journal's Aims and Scope.
- However, a related study appears to have been overlooked: the Wasserstein GAN with Gradient Penalty method proposed by Ishaan et al. (NIPS 2017). The authors are encouraged to discuss this method and, if feasible, compare their proposed approach to it.

Validity of the findings

- As mentioned above, including additional comparisons and discussions would enhance the paper.

Additional comments

- Double-check the reference list, as some entries are missing required information. Additionally, terms like "gan" should be correctly capitalized as "GAN", etc.

Cite this review as

Reviewer 3 ·

Basic reporting

The paper presents a deep generative approach using Generative Adversarial Networks (GANs) to produce customizable patterns for traditional lantern designs. The model integrates classical aesthetics with modern design features, achieving significant pattern diversity and quality, as demonstrated by an Inception Score of 5.259. The dataset, consisting of over 17k images across ten categories, provides a rich foundation for training. The model's architecture leverages noise vector hybridization and post-processing techniques for pattern customization, allowing precise control while preserving cultural authenticity. Comparisons with other GAN-based models like WGAN and LSGAN highlight superior performance. This approach bridges the gap between traditional artistry and contemporary design, modernizing cultural products while retaining heritage elements. The findings suggest practical applications in decorative and functional domains. As great contributions of authors in this paper, I thought it is suitable for publication consideration. Despite these achievements, the method faces challenges like mode collapse and data dependency, indicating room for future improvement. I have some revision recommendations for authors to improve their paper’s quality before it is accepted for publication, including:
1) The abstract is informative but could better emphasize the real-world applicability of the method in broader design industries beyond lanterns.
2) While the introduction outlines the relevance of patterns, it lacks a clear explanation of the unique contributions of the paper. Adding a more explicit comparison with existing approaches would clarify novelty.
3) The section could better articulate how the proposed model addresses existing limitations like mode collapse or interpretability challenges in GAN-based designs.
4) The discussion of related works is comprehensive but does not sufficiently evaluate the limitations of recent methods like WGAN or genetic algorithms in integrating cultural authenticity. Additionally, it is better to include more critiques of competing methods (e.g., computational inefficiencies or aesthetic limitations) would strengthen the justification for the proposed approach.

Experimental design

No comment.

Validity of the findings

Please refer to my basic reporting section.

Additional comments

No comment.

Cite this review as

---

## Round 0.2 · accepted · Accept

The authors have thoroughly addressed all the comments provided by the three reviewers. In my opinion, the manuscript is now ready for publication.

Reviewer 1 ·

Basic reporting

- The introduction and results sections have been refined for clarity and readability.
- The background on Wasserstein GANs has been expanded. Figures and tables have been improved in resolution, labeling, and detail.

Experimental design

- The revised manuscript provides detailed descriptions of the GAN training process, including hyperparameters and evaluation metrics, improving reproducibility.

Validity of the findings

- The discussion has been strengthened by addressing potential biases, scalability, and broader applications of the proposed approach.
- Conclusions are well-supported by the results and address the study’s significance.

Cite this review as

Reviewer 2 ·

Basic reporting

The revised manuscript presents a certain level of professionalism with clear and precise English, good literature references, and sufficient field context, supported by a well-structured format, relevant figures, tables, and shared code & link to raw data, while presenting self-contained results aligned with hypotheses and formal outcomes.

Experimental design

This study aligns well with the journal's Aims and Scope, presenting a clearly defined, relevant, and meaningful research question that addresses an identified knowledge gap, supported by a rigorous investigation conducted to high technical and ethical standards, with methods described in sufficient detail to ensure reproducibility.

Validity of the findings

The revisions show clear links to the original research question. Code and data are provided. Comparisons with other studies were performed well. Rationale and benefit to the literature are explicitly stated.

Additional comments

The authors have thoroughly addressed all of my comments in this revision. I have no additional feedback and recommend accepting the paper.

Cite this review as

Reviewer 3 ·

Basic reporting

After carefully considered all revisions have been conducted within the latest revised manuscript version as well as responses of authors for reviewers’ queries, I confirmed that all problems within the previous manuscript have been resolved. As a result, I thought this paper could be accepted for publication in this form. Thanks.

Experimental design

No comment.

Validity of the findings

Please refer to my basic reporting section.

Additional comments

No comment.

Cite this review as